# Multiscale Computation and Dynamic Attention in Biological and Artificial Intelligence

**DOI:** 10.3390/brainsci10060396

**Published:** 2020-06-20

**Authors:** Ryan Paul Badman, Thomas Trenholm Hills, Rei Akaishi

**Affiliations:** 1Center for Brain Science, RIKEN, Saitama 351-0198, Japan; 2Department of Psychology, University of Warwick, Coventry CV4 7AL, UK; T.T.Hills@warwick.ac.uk

**Keywords:** artificial intelligence (AI), decision making, attention, multiscale computation, environmental neuroscience, prefrontal cortex, exploration-exploitation, information search

## Abstract

Biological and artificial intelligence (AI) are often defined by their capacity to achieve a hierarchy of short-term and long-term goals that require incorporating information over time and space at both local and global scales. More advanced forms of this capacity involve the adaptive modulation of integration across scales, which resolve computational inefficiency and explore-exploit dilemmas at the same time. Research in neuroscience and AI have both made progress towards understanding architectures that achieve this. Insight into biological computations come from phenomena such as decision inertia, habit formation, information search, risky choices and foraging. Across these domains, the brain is equipped with mechanisms (such as the dorsal anterior cingulate and dorsolateral prefrontal cortex) that can represent and modulate across scales, both with top-down control processes and by local to global consolidation as information progresses from sensory to prefrontal areas. Paralleling these biological architectures, progress in AI is marked by innovations in dynamic multiscale modulation, moving from recurrent and convolutional neural networks—with fixed scalings—to attention, transformers, dynamic convolutions, and consciousness priors—which modulate scale to input and increase scale breadth. The use and development of these multiscale innovations in robotic agents, game AI, and natural language processing (NLP) are pushing the boundaries of AI achievements. By juxtaposing biological and artificial intelligence, the present work underscores the critical importance of multiscale processing to general intelligence, as well as highlighting innovations and differences between the future of biological and artificial intelligence.

## 1. Introduction

Recent work suggests that the brain’s computational capacities act over multiple spatial and temporal scales and that understanding this multiscale ‘attention’ (broadly defined) is critical for explaining human behavior in complex real world environments [1,2,3,4,5,6]. Trade-offs between choosing a familiar restaurant versus trying a new one (exploration versus exploitation) or spending money now versus saving it with interest for later (intertemporal choice) are both common examples where multiscale processing can allow individuals to evaluate the consequences of alternative courses of action at different spatial and temporal scales and choose accordingly [7,8]. Indeed, one can reasonably argue that many of the challenges we face, both individually and collectively, are problems that require trading-off the value of information and resources over different scales. This is a problem that is central to both biological and artificial intelligence (AI). Though AI has historically been forced to focus on more narrow single-goal computational architectures [3,9,10,11], much of the progress made in the latest “AI spring” are, as we describe below, achievements of multiscale processing. Such advances are allowing AI to adaptively generalize across contexts of time and space, to increasingly satisfy what a recent definition of AI intelligence characterized as “an agent’s ability to achieve goals in a wide range of environments” [12]. Further progress in developing and understanding multiscale processes is recognized as necessary for developing efficient and adaptable AI systems [9,13,14,15], but it is of course also central to understanding biological intelligence.

To better understand multiscale processing, researchers must answer a number of questions. How general are problems in multiscale processing across tasks and domains? What evidence is there that biological and artificial intelligence benefit from multiscale processing? What architectures best achieve multiscale processing in biological and artificial intelligence? How can these architectures move from fixed to adaptive multiscale intelligence? What environmental conditions benefit most from multiscale processing? Moreover, how can we develop predictive behavioral models using a more comprehensive and generalizable multiscale theoretical framework of biological cognition and AI [16,17]? The requirement to integrate information over spatial and temporal scales in a wide variety of environments would seem to be a common feature underlying intelligent systems, and one whose performance has a profound impact on behavior [16,17,18,19,20,21]. Our focus here is on addressing the above questions by demonstrating the generality of multiscale processing in biological and artificial intelligence and, by juxtaposing them, gaining insight into the architectural advances that underlie innovations in multiscale intelligence more generally.

A key insight is that biological cognitive systems often handle multiscale problems by adaptively modulating ‘attention’ across multiple scales (through mechanisms which can include variable learning parameters or the degree to which working memory is considered) [22,23,24,25]. Thus, intelligent systems do not just set a simple weighting (e.g., Gaussian or exponential) across temporal and spatial scales, but rather, adaptively modulate that weighting in light of circumstance. An overarching analogy is foraging, where an animal must decide how to move in relation to a given resource distribution (Figure 1). In a typical foraging problem, an animal must both consider the opportunities under its nose, as well as the larger global context in which it finds itself. This global context contains a variety of scales and associated distributions: where are other resources, how plentiful are they, with what probability do they persist, how does access to them change with time of day, season, competitors, predators, and so on. A popular solution across animal species is called area-restricted search, which modulates the animal’s movement through space, from global to local, in direct response to resource encounters. In other words, area-restricted search adapts the scale of an animal’s attention in response to past payoffs [8,26]. This strategy generalizes across numerous situations which call for similarly adaptive multiscale solutions. For example, in a multi-armed bandit problem, where an individual must choose among options which vary in their payoffs over time, individuals must assess choices by integrating payoffs over repeated samples, evaluate potential temporal trends in payoff distributions, and explore alternatives appropriately, in order to exploit them optimally in light of time horizons, risk tolerance, and so on [27]. Analogous to area-restricted search, solutions similar to win-stay lose-shift are common in multi-armed bandit problems [28,29,30]. Other examples of the generality of multiscale processing include developing general problem solving and information search strategies [31,32], designing versatile AI systems [9], planning efficient business management strategies [33], and establishing a comprehensive understanding of mental illness [34]. In these challenging real world environments, adaptive agents have to evaluate the long-term costs of current solutions, find and evaluate alternative solutions, and vary the scale over which these solutions are evaluated in light of the current context [31].

In practice, research on multiscale processing is difficult, because many experimental paradigms focus on short spatial or temporal scales. In the studies of spatial processing, the experimenters may assume that the animals do not process information beyond the confines of the experimentally designed space. These experimental constraints and accompanying assumptions are due to complexity issues for researchers, including creating experiments with long enough duration to imitate real world long-term decision making, or overcoming challenges in developing models that encompass multiple spatial scales. Yet, expanding the spatio-temporal breadth of experiments often reveals multiscale capacities that were not previously expected. For example, grid cells were found by expanding the spatial scale beyond that previously used in studies of place cells (i.e., moving rats to larger size arenas for exploration) [36,37,38].

Studies on temporal processing have similar constraints, but also have revealed neural correlates of temporal scale. For example, multiple temporal scales are encoded in the firing patterns of neurons [23,39,40], and temporal properties of individual neurons are related to their specific functional roles in working memory tasks [41]. In the designs of behavioral tasks, a “trial” has been used as the basic unit of temporal organization of the experiments. It has been historically assumed that the information processing in each trial is independent from information processing from other trials, and that once one trial completes, all the information processing is reset [42,43] (e.g., such as in the commonly used drift-diffusion models used to explain processes of perceptual and reward-guided decision making [42,43,44]). However, sequence effects and path-dependence are common in behavioral science, with even studies of perceptual discrimination revealing carry-over effects from the prior trials [45,46,47,48]. In widely reported behaviors in behavioral economics, people’s choices tend to show temporal dependencies across multiple time-scales [49]. While these behaviors have been previously regarded merely as “anomalies” [50,51], it may be more natural to think that multiscale processing has evolutionary significance beyond the narrowly-defined rational theories of behavior that assume an independence between trials that is unlikely in the real world [52,53,54,55].

In what follows, we will explore the thesis that multiscale processing is central to biological and artificial intelligence. We argue that, in fact, a multiscale architecture that can effectively modulate between local tasks while also considering multiple global goals and contextual factors is necessary for agents to perform well in dynamic and uncertain environments approaching real world complexity. We believe that both fields stand to gain from a common understanding of this problem. To do this, we first focus on experimental and theoretical investigations aimed at understanding multiscale processing in the brain and biological systems more generally [35,56,57,58,59,60,61,62,63,64,65,66]. We start with temporal processing and then move to spatial processing, before tackling the neural underpinnings. In the second section, we focus on multiscale AI. There, we will first explore the many generalist multiscale architectures that have given rise to AI’s recent achievements. Then, we will focus on in situ environments in robotics and game AI, where AI is currently facing some of its most challenging applications. Finally, we close by exploring some recent advances at the interface of neuroscience and AI.

## 2. Bias, Inertia, and Habit Formation in Local Environments

In stable or slowly changing environments, where the recent past reliably predicts the future, it is computationally efficient to narrow down the range of options at each decision point. This can enhance efficiency by increasing decision speed and reducing computational costs. Put simply, if the best choice of ice cream in the recent past was vanilla, then chances are it is the best choice now, and you can forego the sampling of other flavors. In this section, we will describe two formal demonstrations of this narrowing of the temporal scale: decision inertia and habit formation.

### 2.1. Decision Inertia

Decision inertia is the influence of previous decisions on current decisions, even when the criteria for decision making should make those decisions independent. The presence of decision inertia suggests multiple temporal scales in decision making [63,65,67]. Akaishi et al. [63] examined participants decisions in a two-direction motion discrimination task and found that subjects made repeated choices across trials, in which a decision in a previous trial biased the decision processes in subsequent trials. Crucially, this bias also accumulated across trials [63] (Figure 2, the left panel). This observation of accumulating bias can be explained by the multiple effective temporal scales of decisions. In other words, the choices made on each trial are not just a choice about the immediate sensory environment of one trial, but also a decision about the states of the environment extending beyond a timescale of one trial. Thus, the temporal scales of neighboring decisions overlap (Figure 2, the right panel).

Another important message that can be derived from the right panels of Figure 2 is that the shorter time scales (upper panel) and longer time scales (lower panel) of information processing are overlapping. Such overlap can be observed in the equations of the best model explaining the behaviors, or decision variable DV related to the choice repetition probability *P* for a subject to make a choice *C* of “left” (*C* = −1) or “right” (*C* = 1) per trial, of subjects [63]:$$CE_{n}=CE_{n-1}+\alpha \left(C_{n-1}-CE_{n-1}\right)\;DV_{n}=\varepsilon M_{n}+\beta CE_{n}\;P_{n}=1/(1+\exp\left(-DV_{n}\right)$$

Abbreviations are *CE* for choice estimate, *C* for choice in each trial *n*, *M* for the motion stimulus presented in each trial, and α, ε, and β as fitting parameters. DV, which influences the ultimate choice, consists of the short-term effects of the motion stimulus in the current trial, plus the influences of the previous choice (which creates a bias towards repetition). Thus, like area-restricted search and some AI architectures that we will discuss later, *CE* contains a memory of past experience that is used to inform present choices.

In summary, decision inertia exemplifies a common finding in biological cognition of using recent information to inform decisions in the present. It is perhaps the simplest form of multiscale processing, but it is ubiquitous. Moreover, in a world with spatio-temporal autocorrelations—where recent and close encounters predict present circumstances—decision inertia may be the best choice when decisions must be made quickly. Moreover, the research described above also demonstrates that this narrowing is adaptive to persistence in the context [63].

### 2.2. Habit Formation

A critical multiscale issue for brains of all species is how to prioritize, allocate attention to, and even automate, the processing and execution of an overwhelming amount of simultaneous sensory information about local tasks, while considering larger global contexts and changing environments. While a detailed analysis of one’s surrounding can lead to more accurate decisions, costs may prohibit an agent from repeating the same analysis each time they enter a similar environment (e.g., walking to one’s workplace routinely). Instead, habit formation occurs when repetitive computations are streamlined, which can greatly increase computational efficiency without the loss of performance in static environments. Habit formation chunks parts of action sequences into an aggregated unit [68,69,70,71,72,73]. Patterns of neural activity across multiple brain areas show corresponding changes, as the behaviors of repeated actions exhibit signs of the habit formations. Reaction times also get shorter [70]. Remarkably, patterns of reaction times also show increasing structural organization: several actions are chunked together and the time intervals between the chunked actions get smaller compared to the intervals of non-chunked actions [68,74]. The increasing behavioral organizations are reflected in the patterns of changes in neural activity in cortical and subcortical brain areas [75,76]. For example, as the habituation process proceeds when monkeys perform repeated sequential movements by touching specific parts of a screen presenting binary choices, the single-unit neural activity in their medial frontal areas gradually shift from the more anterior areas (pre-supplementary motor areas) to the more posterior areas (supplementary motor areas) [75,77]. Similar shifts of anterior to posterior brain areas have been observed in fMRI activity patterns as well [78]. These patterns of the behavioral and neural activity are summarized in the theories of hierarchical reinforcement learning [69].

What causes these patterns of habit formation is still debated. Some researchers claim that these habit formation processes occur as a result of reward-guided learning [79]. Still other researchers posit that reward is not necessary for habit formation and instead habits are the result of a bias towards repeating previous choices, even if that choice no longer provides value [59]. There are likely multiple mechanisms driving habit formation, such as a drive towards repeating easier tasks to minimize mental effort [6,80], as well as multiple types of both maladaptive and adaptive habits rather than just one category of habits [81]. Nevertheless, the situations examined so far suggest a critical role of uncertainty in decision processes. Redish and his colleagues examined the deliberation-like behavioral and neural activity pattern in rats performing modified T-maze tasks [82,83]. In the analysis of behavior, the deliberation-like behavior at the decision point of the T-maze decreases as the learning proceeds in the same experimental condition. Consistent with this behavioral change, the neural activity in the hippocampus and related brain areas progressively show fewer patterns of deliberation-related neural activity (i.e., less mental wandering around the spatial locations between the two arms of the branching point). Instead of always considering each option with the equal distribution of time, the neural and behavioral patterns progressively show more fixed patterns and the exploration of fewer alternatives. Thus, this focusing of the mental processes on a fixed course of action makes the computation more efficient if the environment does not change drastically, but at the risk of future bias if the environment starts becoming more dynamic. However, whether such focusing of mental processes results from primarily from mentally “replaying” recent choices (choice bias) more frequently, or replaying the best calculated choices (value-guided) more frequently, remains an interesting topic of future research [84] (replay is discussed in more detail below).

## 3. Multiscale Decision Making in the Global Environment

The tendency of making computations more efficient in the local environment is a double-edged sword. Local adaptation, especially the narrowing down of the computational scope, is problematic in terms of opportunity cost. That is, with the narrowly tuned optimization processes, the animal in the environment can miss the crucial resources that could be available if the animal maintains larger scale computations about the wider environment (Figure 1). Therefore, it is highly likely that animals, including humans, are equipped with the capacity to counteract the tendency toward local scale computations, which is reflected in the inertia and habit formation described above. The movement of the research community to understand such global computation has a long history [8,26,83,85,86]. One research area that examined decisions of multiple time scales is foraging theory [87].

### Foraging Decisions

Foraging decisions routinely require trading-off short-term opportunities for exploitation against long-term opportunities for exploration. In many cases, exploration is a shorthand for the expected time allocated to move to or find a new resource patch. Multiscale optimization is necessary because, in the short-term, the gains that could be harvested from the present patch are greater than the gains that will be available during travel. If an individual is a greedy optimizer, incapable of optimizing over longer time scales, they will stay in a patch until the rate of resource acquisition is equal to or less than the rate that would be acquired during a travel period. Most animals leave patches long before this.

The solution to this multiscale optimization problem is captured well by the marginal value theorem (MVT) (Charnov 1976) [88]. For the MVT, an animal is assumed to forage in a world of resource patches, with an infinite time horizon, and one simple decision: when to leave a patch. Foraging in a patch depletes its resources, leading to a reduction in the resource intake rate, but traveling to a new patch costs time, during which no new resources are acquired. Charnov (1976) showed that the solution to this problem was
$${R\left(t\right)}^{\prime }=\frac{R\left(t\right)}{t+T}$$
where the instantaneous rate of resource intake at a patch, *R*(*t*)*′*, equals the long-term average rate across all patches, where *R*(*t*) is the cumulative resources acquired in each patch, *t* is the time spent foraging in that patch, and *T* is the time spent traveling between patches. Many animals show qualitative consistency with this model, for example, by staying in patches longer when the travel time between patches increases [83]. However, they nonetheless frequently leave the patch long before it is depleted in favor of higher average long-term gains.

Using operant scheduling paradigms, psychologists have demonstrated this in the laboratory by giving animals the choice between two levers, one with a progressive-interval schedule (in which the time or presses between rewards increases), and one with a relatively long fixed-interval schedule (time or presses are fixed). However, in this paradigm, a fixed-interval choice also resets the progressive-interval choice, much like a travel-time between patches. This work has shown that animals such as pigeons and chimpanzees choose the fixed-interval schedule, even when the progressive-interval schedule would pay off sooner [89,90,91], resetting the progressive-interval near the point predicted by the MVT. Similar optimal strategies have been found for human memory foraging [86] and information search [92].

Studies of foraging decisions in neuroscience have helped to enhance our understanding of the neural processes that might guide this trade-off [64,87,93,94]. The hallmark of such foraging tasks is the superimposition of computations at the shorter time scale (a trial) and the longer time scale (a block of trials). For example, in the study by Kolling et al., subjects demonstrated a capacity to dynamically modulate choice between safe and risky alternatives, in response to a changing global context [64]. Over a fixed number of trials, subjects were tasked with making a decision between high risk/high reward or low risk/low reward options. The subjects were also required to reach a certain threshold of accumulated reward within a block of trials. If the threshold was not reached, all the accumulated rewards were removed, and the subjects received nothing. If the amount of accumulated rewards from the outcomes of a series of risky decisions was far away from the target threshold toward the end of the block, the subjects felt pressure to make even more risky decisions, which had a higher reward magnitude, but a lower probability of success. Kolling et al. demonstrated that this conflict between short-term (safe options) and long-term (risky options) was mediated by the dorsal anterior cingulate cortex (dACC), which we discuss further below. The tendency to take a risky choice has been regarded as a stable trait of an individual, but the study results showed that contextual global factors can rapidly modulate the risk-aversion tendencies of human participants in a dynamical manner.

In another experiment demonstrating multiscale decision making, subjects performed two tasks [66] (Figure 3). One task was to push a button and receive an outcome based on a predetermined schedule of reward rates (Figure 3a). For example, in the schedule of decreasing reward rates, the reward amount and its probability of occurrence were high in the beginning of the task block and later decreased. In the schedule of increasing reward rates, the opposite trend was true. The second task was a patch leaving decision, in which subjects decided whether to stay in the current environment (patch) or leave to the other environment (patch) (Figure 3e). The subjects’ valuation of the prospects of their current patch compared to the value of leaving involved the interpolation about the trend of reward rates in the future, which were represented simultaneously in the dACC. Comparisons with a simple reinforcement learning model (RL-simple) demonstrated that human participants made decisions based on the instantaneous reward rate and the reward rate trend. The RL-simple model failed to do this, “integrat[ing] historical and recent rewards into a single value” [66]. When performance depended on the reward rate trend, humans substantially outperformed RL-simple. Better models showed that the leaving decision was programmed to land the subjects in the patch whose reward rates were between the good (increasing) and the bad (decreasing) patch (Figure 3a, the line of default patch in red). The reward experiences in short-term periods in the immediately preceding past showed the positive influences favoring staying decisions and the long-term aggregate reward experiences in the total past periods showed negative weights favoring leaving decisions (Figure 4). This negative weight in the long-term aggregate may seem odd, especially considering that the reward experiences in the distant past in the current patch reinforce the decision to leave the current patch. This may reflect the subject’s knowledge of the longer scale task structure, in which the initial rich reward patches became poor patches, and vice versa. Alternatively, in the known phenomena of contrast effect, the recent reward experiences are interpreted in the context of the long-term reward averages. Thus, the apparent negative effects of the distant reward experiences may not reflect the simple local negative influences of these experiences but the consequences of the multiscale computations, in which local events and global contexts are compared.

In a related study, Constantino and Daw found that multiscale processing also explains foraging behavior better than standard models of reinforcement learning [95]. Empirical comparisons between a model with long-term temporal scales based on the marginal value theorem and a model with short-term temporal scales (temporal-difference learning), a very similar RL-simple-like model to that in Figure 4, found that accounting for additional long-term temporal scales explained subjects’ behavior better (Figure 5) [95].

Similar promising results of model comparison in multiscale computation were also obtained in the previously discussed study by Wittmann et al. [66], and additional studies in recent years [56,57,58,59,60,61,62]. Thus, mounting evidence suggests that the typical approach to information processing in humans is multiscale processing. For example, even a semantic memory search in category fluency tasks resembles optimal multiscale strategies for foraging in a patchy environment, following the predictions of the marginal value theorem [86].

An important future goal is to create multiscale neural computational models that better predict more complex real world behaviors [96] (many of which can be directly understood through the general foraging paradigm [8]). Inspiration for multiple temporal or spatial scale computational models may also be found in the physical and mathematical sciences. Certain search algorithms that simultaneously operate over multiple scales can be highly efficient at solving complex problems, including the previously discussed foraging situations, where animals need to explore both the area around local patches to properly assess each patch value, as well as explore over long distances for (potentially) higher value patches. As noted above, the evidence suggests that most animals do this by engaging in a process called area-restricted search, with local search in response to recent resource encounters, and only giving up slowly as encounters with resources diminish [26,97]. In contrast to Lévy flights, which are multiscale only by sampling from a fixed distribution of travel distances over multiple scales [26,98,99,100], area-restricted search adapts the spatiotemporal search scale in response to context, much like the attention processes that we will see later in the Multiscale AI section. Though the multiscale property of Lévy flights have been shown to outperform Brownian motion [26,101,102], the adaptive multiscale search in area-restricted search outperforms Lévy flights, and can also produce power law distributed search path lengths [103,104,105].

## 4. Neuronal Bases of Multiscale Computations

### 4.1. Dorsal and Ventral Medial Prefrontal Cortex

What neuroanatomical regions are most critical in modulating the interplay between local and global computations? There have been several important experimental results providing insight for this question in foraging research, by differentiating between “within-state” (local) and “state-change” (global) decisions [106]. In within-state decisions such as many economic decisions, the agent makes choices based on relative values of the immediate options presented to them. Within-state decisions are by far the most commonly studied in decision neuroscience [107,108]. In global state-change decisions, the agent is computing the total value of switching to a new environment (or different global state), based on a cognitive map of probabilistic relationships between stimuli, actions, and outcomes created from previous experiences [106]. As briefly described above, several researchers have found adaptive global state-change decision-making—a clear sign of multiscale processing—to be strongly related to activity in the anterior cingulate cortex (ACC), the neighboring dorsomedial prefrontal cortex (dmPFC), and often brain subregions directly dorsal or anterior to the ACC (e.g., the anterior medial prefrontal cortex or the pre-supplementary motor area) [106].

The ACC has multiple attributes that make it suitable for performing global computations, including sensitivity to both current rewards and trends in reward [66], sensitivity to the rate of change of environmental variables [22], very long integration time constant neurons for tracking slow changes in reward rate [40], and the capacity for tracking multiple timescales simultaneously [23,106,109,110]. The involvement of the dmPFC is also expected to play a role in global decision making, since it is generally active when agents explore environments, and is connected to both subcortical reward centers and motor circuits [106]. In addition, the orbital frontal cortex (OFC) has been shown to track an agent’s previous states, which may allow a comparison between previously sampled local environments for combined local-global decision making [110,111,112,113].

### 4.2. Lateral Prefrontal Cortex

The dorsolateral part of the prefrontal cortex (PFC) plays important roles in modulating the processes occurring in other parts of the brain through extensive anatomical connections [114,115,116,117]. By transmitting top-down signals, often associated with mechanisms of attention or other executive functions, the global representation of the agent’s task context influences the more local processing of stimuli or simple motor actions [118]. The importance of such hierarchical information processing for managing mental resources and connecting multiple scales of computation has been recognized, but mapping the full architectural and theoretical framework behind the PFC’s higher level executive control and the modulation of lower level processes across the brain has remained elusive [119]. The recent interest in artificial analogues to this top-down attention and executive processes in AI, especially research about multiscale architectures for neural networks, is providing a renewed interest in the neuroanatomical substrates of top-down modulations within hierarchical and multiscale computations [119,120]. Note that such biological top-down attention is distinctly different from (but complementary to) the bottom-up attention processes, based in the temporoparietal cortex and inferior frontal cortex [121], which are critical for tracking and prioritizing new stimuli in dynamic environments and may have multiscale relevance, especially for real world autonomous robots [122].

Generally, further understanding of how and why such biological architectural organization provides increased computational efficiency, power and adaptability from the perspective of computational algorithms are critical questions for the neuroscience community to address in the next decade. Inspiration from the AI community will facilitate the understanding of this hierarchical/multiscale organization, especially as more intricate artificial neural network structures are explored to improve the performance of applications in dynamic environments [13].

### 4.3. Cellular Mechanisms

There has been increasing recognition of the importance of neural representations of multiple temporal scales in memory and cognition. The influential work by Xiao-Jing Wang and colleagues reported analyses of single unit activity in cortical areas of monkeys performing behavioral tasks (Figure 6) [23,40]. This work demonstrated a hierarchical ordering in the timescales of temporal autocorrelation in cortical firing. With sensory areas exhibiting shorter timescales of temporal autocorrelation, parietal areas showing intermediate timescales, and prefrontal areas showing the longest timescales, which ranged from milliseconds (sensory areas) to seconds (prefrontal areas). This maintenance of information over various timescales is useful for tracking trends over longer periods of time, as well as rapid updating and responses.

The neuronal basis of multiple scales of computation has been considered in memory research as well [123]. The discovery of place cells and cells with grid-like patterns [37] provides a promising framework, allowing an agent to perform detailed computations at both global and local scales for complex tasks [16,17,124,125]. This is analogous to the time scales in the foraging paradigms discussed in previous sections, where local and global information must be incorporated into optimal behavioral accounting [126]. Additionally, researchers have observed that, after an exploration or short-term foraging event, a “replay”, or reenactment of the neural activity patterns that occurred during the learning event, can occur in the hippocampus and cortical areas (especially the entorhinal cortex), that may allow for credit assignment computations [4]. Some evidence exists that these grid cell capabilities could be useful during tasks that require multiscale computation, such as calculating shortcuts from previous movement trajectories [84,127], predicting the attributes of unexplored regions (i.e., evidence has been observed for “preplay” in rats [128], a possibly critical feature of multiscale computation), and performing future planning during goal-directed behavior [82,129,130].

Promisingly, grid cell behavior has been observed directly in humans with single-neuron scale electrode data [131], consistent with indirect fMRI observations of grid-like fMRI response patterns in humans, during both navigation and conceptual thinking [125,132]. Significant numbers of grid-like cells (and place cells) were observed in multiple brain regions, with most in the entorhinal cortex, hippocampus, and cingulate cortex [131]. Thus, there is potential for future experiments to expand the roles of grid cell computation to other species and additional non-movement related behaviors, with several research groups recently putting forth detailed computational models for how grid cells and place cells may interact to create flexible cognitive maps containing predictive representations (e.g., multiscale “successor” representations) of combined spatial information, valuation, and other conceptual knowledge [16,17,133,134]. A critical goal for future researchers in this field is to design experimental frameworks to test such models in a laboratory.

## 5. Multiscale AI

As noted by Chollet in a recent article on machine intelligence, “The promise of the field of AI, spelled out explicitly at its inception in the 1950s and repeated countless times since, is to develop machines that possess intelligence comparable to that of humans” (p. 3, [135]). One of Chollett’s key observations is that intelligence is inherently a problem of scope. Ambitions for *general* AI, comparable to human intelligence, are therefore ambitions for intelligent systems that can meaningfully approach the scope (i.e., breadth) of human-relevant tasks. This is inherently an aspiration for a multiscale intelligence. Such an intelligence has both the generalizable adaptability to quickly learn local (constrained) tasks (e.g., similar to how humans within days or weeks can learn cooking, driving, swimming, etc.), while also being able to consider the global applicability of these tasks in a wide range of dynamic environments [135,136]. One mid-range example of this scope is the motivational capacities that allow for active learning, whereby an AI can actively sample and construct its own training data [137,138]. In what follows, we discuss several key examples of progress and challenges in multiscale processing in AI. This is not designed to be a thorough review for knowledgeable AI practitioners, but a discussion of several examples that demonstrate similarities and differences between biological and AI multiscale processing.

Substantial progress has been made towards developing multiscale AI over both temporal and spatial scales. Temporal scale examples include continuous-time reinforcement learning models [139] and continuous time Bayesian networks [140]. The latter, for example, can be used to model the temporal characteristics of drug effects and their likely consequences on drowsiness and pain in the context of changing stomach contents. Spatial scale examples include visual object recognition [141] and natural language processing [142], with the latter facing multiscale challenges such as how to interpret current text based on past text. Such approaches have had success in numerous applications, including cybersecurity [143], genetic and clinical analyses [144,145], and robotic exploration with real-time learning during continuous actions [146].

Taking a step backwards, in many respects, multiscale processing is the fundamental architectural advance that has led to the current “AI spring”. Which is to say, much (perhaps all) of deep learning is focused around using multiple processing layers, to achieve behavior that can harness predictions at multiple scales of abstraction [147]. The fundamental multiscale workhorses in this area are recurrent neural networks and convolutional neural networks. Recurrent neural networks (like long short-term memory networks, LSTM) learn and summarize context information by updating a hidden state (either explicitly or implicitly, respectively), in a sequential step-by-step process. This allows them to carry information from one time-step to the next, allowing the algorithm to build up contextual representations based on sequential experience, such as one might have when reading text. Deep convolutional neural networks, on the other hand, piece together information at multiple scales, by passing information through filters (convolution kernels), which are tuned during training and can learn local patterns in the input feature space. By passing information through kernels of possibly varying size, deep convolutional neural networks can learn spatial hierarchies representing increasingly abstract visual relationships between information at different scales (such as noses on faces on heads in a crowd). Convolutional neural networks make multiscale processing particularly apparent because scale information can be used in both directions, to categorize entire images as well as to provide pixel level classifications within the image [141].

More recent advances in multiscale AI involve the capacity to adaptively modulate the scope of the context, much like the area-restricted search foraging strategy described in the introduction. Perhaps the most successful of these approaches is based on a process called attention [148]. Attention draws global dependencies between information at various scales, in order to compute a representation of the relevant context around a particular input value. For example, in machine translation, the problem might be to translate the French sentence *“tremble comme une feuille”*, into English. To do so, an attentional context vector is computed, that weights neighboring words in relation to their relevance to the current word in the output translation [149]. As this context is likely to change depending on the word, attention provides an efficient mechanism for summarizing relevant information across an entire input sequence without having to focus on everything. Because attention is dynamic, this allows the decoder of an encoded semantic vector to vary the scale over which its attention operates, allowing it to distribute its attention in relation to the current word, sometimes called self-attention. For example, in our French sentence above, the word *feuille* can mean either *leaf* or *sheet (of paper)*. However, by attending to the word *‘tremble’*, the decoder is more likely to translate *feuille* as the English word *leaf*, producing *tremble like a leaf.* Similar approaches have also had success in captioning images, by focusing attention on various regions of an image during the decoding process [150].

The capacity to modulate attention has also been introduced to convolutional networks in the form of dynamic convolutions, with dramatic success [151]. Dynamic convolutions use adaptive convolution kernels at each time-step. Instead of fixing the convolution kernels after the initial training, the kernel weights are evaluated with respect to the current input data. This again allows for multiscale processing that adaptively constrains information processing in light of the present context.

A promising extension of attention are transformer networks involving multi-head attention, which allows algorithms to attend to information over multiple contexts simultaneously, that is, in parallel [148]. For example, in machine translation, multi-head attention produces multiple attention vectors for each word, analogous to simultaneously answering questions, like who, what, why, where, and how, for every word in the translation. By altering the relative tuning of this multiscale information, transformers using self-attention mechanisms can process all the words in the input simultaneously, computing contextual information at multiple scales, and harnessing this high-dimensional multiscale information to produce state-of-the-art performance in natural language processing, such as machine translation and question answering (such as GPT-2 and BERT, [142,152]).

Multiscale AI is likely to continue to benefit from neuroscience-inspired AI [9], such as processes analogous to human executive functions and the PFC, which have been proposed to maintain goal representations in a hierarchical fashion, allowing for multiscale goal representations [114,120]. Similar inspiration may come from other higher level computational processes, such as working memory-like conscious binding (e.g., a “consciousness prior”) [153]. In each case, the scope of the multiscale process is expanded.

Nonetheless, despite the above advances, efficient multiscale processing has remained a longstanding challenge in real world environments, especially where there are high degrees of uncertainty and multiple contextual scales [154]. Surprisingly few AI algorithms are available for real-time learning in such environments [146]. This challenge is becoming even more urgent, as the expansion of AI continues to unfold and the complexity and scope of the tasks, and their associated data sets, increase (e.g., self-driving cars, medical diagnoses, scientific research, etc.) [135,155].

Thus, while AI systems can outperform humans in many tasks [13,14,135], the current approach to achieving this high performance often results in or requires overspecialization, leading to failures to generalize to novel or overly-dynamic environments [156]. This approach results in AI agents still significantly underperforming, relative to humans, as their environments and assigned tasks approach the complexity and rich dynamics of the real world (e.g., navigating unexplored regions, solving tasks which require communication, tracking and prioritizing multiple goals or sub-goals, etc.) [13,130,156]. Thus, a further understanding of how humans perform so well in these more complex environments, even in the face of sparse data, is a crucial factor in resolving the performance gap between human and AI agents [9]. From the neuroscience side, attaining this knowledge will require designing experiments that both better approach the complexity of the real world and require multiscale computations to complete [64] (Figure 7).

In the next sub-sections, we discuss two key applications of multiscale AI that provide a grounding for the more abstract multiscale neural networks architectures described above: progress towards generalizable, autonomous AI in (1) robotics and (2) electronic video games.

### 5.1. Autonomous, Generalizable Robotic Agents for Real World Environments

Moving towards such generalizable, autonomous human-like or super-human intelligence is not just an academic exercise for achieving the sentient AI agents seen in science fiction, there are already many real-world applications requiring such multiscale generalizable intelligence. For example, NASA has explicitly called for such generalized systems [154]. Mars rovers need to achieve a changing list of scientific experiments in finite time—experiments, individually assigned different scientific values—that each require different temperatures, travel times and thus solar power, physical orientation and positioning of the rover, identifying specific environmental features of interest (whose existence may not be known a priori) that could retroactively alter the scientific value of certain proposed experiments, etc., all while the highly dynamic and unpredictable Martian weather system alters the immediate environment and lowers or raises the feasibility of certain experiments [154,157]. Similar complex multiscale needs are present in a variety of robotic AI applications, including underwater scientific exploration [158], search and rescue [159], and even house cleaning [160].

Importantly, most current AI deliberation and traditional contingency planning techniques assume only a small set of discrete outcomes are possible, which leads to severe underperformance in uncertain environments with continuous time-space or other continuous parameters (e.g., power consumption over uneven terrain), and thus do not scale well to larger global problems where all possible outcomes cannot be listed [136,146,154,161]. As a result, modern AI agents primarily outperform humans only in controlled industrial environments, and run into severe challenges in the real world applications discussed previously [162]. Recent work has surveyed, in detail, the progress and performance of different multiscale deliberative-explorative algorithms intended for furthering the goal of autonomous AI in uncertain environments, concluding that algorithmic progress towards robust generalized AI is promising in a modular and context-dependent way, but overall is fragmented and needs to be incorporated into a more global and generalized framework [135,136,163]. Most historical work has been based on model-free RL (more similar to the biological habituation or at least local task learning discussed above [5]), due to the ease of implementation and not requiring prior knowledge to be encoded [164]. However due to the increasingly poor performance of model-free algorithms as the complexity of dynamic environments increases, model-based algorithms have been closely revisited in recent years (though largely in self-designed controlled environments [164]). Model-based algorithms represent a model of the causal structure of their environment, which allow for more complex relationships than model-free algorithms. As a result, model-based algorithms greatly increase sample efficiency, speed up optimal solution convergence, reduce environmental interaction (and thus reduce wear-and-tear damage on the robot), and improve multiple goal tracking in dynamic environments [164,165]. However, learning accurate models, especially in complex environments, has remained a serious challenge, and several recent works have cited lack of reproducibility and open source code in this area, as a major structural bottleneck for accurate benchmarking of progress in this area [164,165,166].

Neuroscience is expected to continue to play a critical role in aiding to enhance the performance and adaptability of both model-free and model-based approaches [69,162]. For example, further interesting robotic AI approaches have tried to implement transfer learning (i.e., a machine learning approach for efficiently transferring knowledge learnt from one problem to another related problem without total retraining, which humans seem to frequently do) in robotic systems for more generalizable human-like learning [167], finding significantly enhanced physical robot performance over traditional deep RL and policy techniques [168,169]. Moving forward, an especially critical interdisciplinary point to resolve is what role higher order human cognitive processes like creativity, curiosity, intrinsic motivation, the creation of cognitive or world maps, etc., as well as bottom-up attentional systems for more adaptable response to dynamic environments [121,122], play in providing our species with such highly generalized intelligence, and whether these processes are necessary or optional for achieving generalized intelligence in AI systems [16,170,171,172]. However, answering such a question requires further understanding of these poorly understood and underexplored processes in human neuroscience as well [171]. 

While exploration of highly dangerous, unpredictable and scientifically rich environments like planetary surfaces and the depths of the ocean, in some ways, represent the ultimate challenge for autonomous AI, a promising proving grounds of more intermediate complexity has also emerged for the goal of developing autonomous AI that can handle uncertain and dynamic continuous time-space environments: electronic video games.

### 5.2. Autonomous Game AI for Quasi-Real World Environments

The multiscale deep neural network algorithms discussed previously are also leading to substantial progress towards generalized AI intelligence in the quasi-real world environments of modern electronic video games, which can provide a more controlled virtual environment to test multiscale AI architecture, without the complications introduced by real world environments and robotics hardware [173]. This area has already attracted considerable interest from social neuroscientists, as these game AI systems are designed to compete (or even cooperate) with humans in modern electronic video games [13,174,175]. Furthermore, modern electronic game AI agent development has partly evolved into an interesting applied human neuroscience problem, due to the unique situation of a large global consumer base putting constant pressure on game developers to implement AI characters and agents in video games that algorithmically behave more like humans, for a more immersive user experience [176], providing an opening for neuroscience input.

Interestingly, this transition in the AI field of moving from simple discretized games to “continuous” (small time scale), electronic real time strategy (RTS) games is mirroring a similar transition in neuroscience, away from using discretized independent-trial experiments [63,177,178] (Figure 7, and see previous sections). RTS games such as Dota 2 have been chosen, as they have continuous-time (~10–100 Hz frame rates) and -space (sub-mm^2^ pixels), thus which may violate Markov assumptions of independent states, and have more dynamic environments than turn-based games such as chess [174,179]. There is a vast difference in the AI computational complexity between these RTS games compared to the simpler turn-based games explored by AI researchers previously (e.g., chess [180], Go [181], backgammon [182]), and even compared to the electronic Atari games played by DeepMind’s deep Q-network agent [183]. In turn-based games, such as chess, the pieces can only move in a small set of well-defined, discrete positions across temporally discrete “rounds” of a game, and players have perfect knowledge of their opponent’s past and current piece positions. In contrast, for the behavior paradigm in these RTS games, agents have to work (with fast reaction times), either alone or in teams, to explore unknown map regions for resource seeking, use the resources to improve their own fitness and their team’s fitness in the face of uncertain and dangerous environments, coordinate global strategies to predict unknown enemy movements and resource locations, and defeat the enemy according to game-dependent metrics for victory [174,179].

**Figure 7 brainsci-10-00396-f007:**
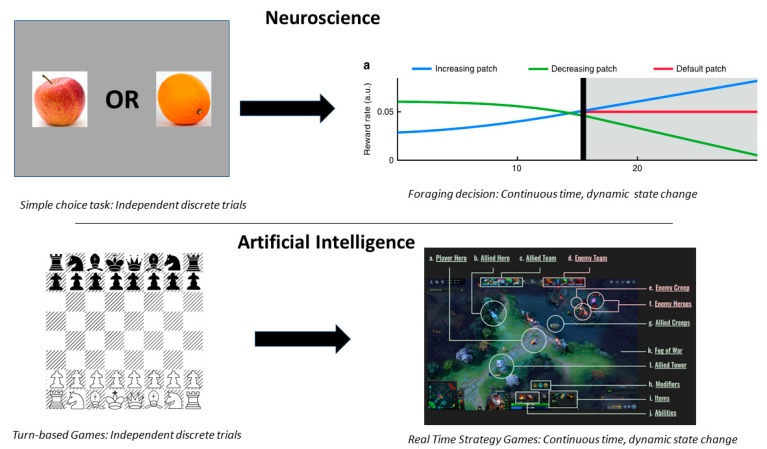
Transition from Independent Discrete Trials to Dynamic Continuous-Time Environments in both AI and Neuroscience. (Top) It has been historically assumed that the information processing in each trial during a neuroscience experiment is independent from information processing in other trials and that once one trial completes, all the information processing is reset. However, in the real world, events and decisions occur over continuous time frames and are often correlated with people’s choices being known to show temporal dependencies across multiple timescales. This has motivated a transition in neuroscience to continuous time experimental designs. (Bottom) A similar transition from spatially-temporally discrete turn-based games to continuous time and space real time strategy games is occurring in the field of AI research, such as from the ancient game of chess (Left) to the modern electronic real time strategy game Dota 2 (Right). Upper right image adapted with permission from Wittmann et al., Nature Communications, 2016; Nature Publishing Group [66]). Lower right image is taken from OpenAI [184], according to the license agreement.

These RTS tasks inherently require multiscale computations for agents to play [185], and thus are a framework that arguably has relevance for behavioral scientists [13,179]. The current RTS games most studied by AI researchers (e.g., Starcraft with several AI systems [186], Dota 2 with OpenAI’s OpenAI Five [184,187]) have elements of traditional foraging behavior. RTS game agents must calculate the risks-versus-rewards of seeking new resource patches in dangerous environments as their current patches dwindle [186], while keeping a long-term goal in mind, similar to the paradigm with global risk pressures discussed previously in Wittman et al. [64] Additionally, multiple scales of computation are needed, often involving transformer networks like those described above, because players are presented with a game display screen that only shows a small local environment contained within a larger game map, where the larger map is only presented as a small symbolic display insert in the main game screen (Figure 7) [174,184,185,186]. Thus, as agents accomplish local immediate goals, they also must simultaneously track global changes in their maps as unknown regions are gradually explored, and new resource patches are suddenly revealed through exploration. Remarkably, some AI agents are now able to reach high performance in multiple different games simultaneously (a strong step towards generalizability), while using only deep reinforcement learning on the same raw pixel input as humans, compared to the highly processed input of older generations of game AIs [183].

Numerous challenges remain in developing AI that can process and modulate attention like humans over multiple spatial and temporal scales [11]. Traditionally, to achieve human performance, AI has relied on local reinforcement learning and search algorithms over massive data sets, combined with superhuman levels of global knowledge [13,14,174]. Such learning strategies are computationally inefficient relative to human cognition, especially at transferring the learned knowledge to other tasks [13,188]. Generally, AI systems require massive retraining if the goal is switched within the same game framework (e.g., survive a round as long as possible instead of score maximization) [13,135,183]. For example, the OpenAI Five Dota 2 player, which defeated human master players in an internationally watched 2019 competition [184], used large-scale reinforcement learning with ~50,000 CPUs plus ~500 GPUs that trained for 45,000 years of game time, and could only perform well inflexibly in very restricted game conditions [135,184]. Thus, the ~10 W human brain seems to be using considerably more efficient multiscale learning and search algorithms to make approximation-motivated (e.g., inference) but accurate decisions in the face of dynamic goals and environments. This decision power is additionally impressive given the context the mismatch in processing times of a human’s ~10 s–100 ms timescale for the fastest perceptual feedforward tasks [189], compared to a computer’s nanosecond scale timescale with typical GHz clocks. Thus, neuroscience-inspired algorithm and structural improvements may more broadly hold substantial potential for the development of more compact and power efficient (i.e., more environmentally friendly) AI computer centers as well.

Recent work has intriguingly found promise in non-reinforcement-learning-based quality diversity (QD) and backplay algorithmic methods, as part of a broader architecture (that can be combined with reinforcement learning) researchers called “Go-Explore” and “Plan, Backplay, Chain Skills” respectively, where agents can revisit useful prior “stepping-stone” environmental states as starting points for further exploration (an ability generally not readily available to physical robotic agents, highlighting the power of the virtual domain) [190,191]. This approach was found to especially improve explorative performance in specific games where reinforcement learning alone fails (e.g., games that require difficult exploration or more global computations) [190,191], and is similar to some foraging behaviors described previously [190]. Collectively, this interesting work suggests that large-scale deep reinforcement learning (which generally only excels during local interactive learning or in controlled environments where complete, and often superhuman [174], global information is provided) is not sufficient for generalized intelligence. Instead, these supplementary “stepping stone” algorithms are hinting at the importance of spatio-temporal (or even conceptual [125]) cognitive map-like algorithms, in achieving more generalized multiscale intelligence that can excel in dynamic, uncontrolled environments [16,134]. Additionally, developing relational inductive priors that are more suitable to the complexity of the real world [192], or more global-thinking hierarchical macro-strategy design, may be needed for significant performance enhancement in more challenging environments [15].

## 6. Discussion

Multiscale AI deep learning algorithms have allowed AI to surpass human performance in many specific tasks in a variety of controlled environments, both virtually and physically, with game-based and robotic AI agents, respectively. While notable progress has been made towards incorporating this local task mastery into more generalized global architecture, generalized intelligence in AI still falls short of the speed and adaptability of human (or even animal) intelligence in dynamic environments (Figure 8). Recent promising biologically realistic theoretical work has provided insight into how the locally efficient choice biases may combine with long time-scale global cognitive maps (e.g., entorhinal grid cells) to provide such efficient, flexible and generalizable multiscale computation in humans [16,17,133,134]. We believe further understanding of this multiscale architecture has the possibility to provide breakthroughs in AI development, similar to neuroscience’s previous contributions of reinforcement learning [193] and artificial neural networks [194,195,196].

Distinguishing between single and multiscale computational processes is still rare in behavioral and neural sciences with regards to exploration and exploitation, but recognition is growing in the importance of considering this topic. Understanding the balance of these multiple competing and/or complementary computational layers underlying multiscale decision making (which may actually be the dominant form of decision making in humans) is a crucial problem for studies of information processing of the brain in the upcoming decade. Moreover, as described above, it is being increasingly incorporated into AI.

One area we have barely touched on is the necessity for intelligent systems to escape local minima in the fitness or value space. For example, imagine the animal in Figure 1 trapped permanently at a suboptimal peak. Current uncertainty reduction models have the weakness of not adequately explaining or predicting what drives humans to break out of habits or biases (such as creating temporally local increases in uncertainty) [197,198,199]. These are drives which presumably include a desire for creating a long-term reduction in global uncertainty. Biological attentional processes may also play a critical role in this area, especially in non-reinforced preference or state changes [200]. Thus, it is an open and important question in multiscale neuroscience as to what motivates humans to embark on risky exploratory behavior by temporarily increasing local uncertainty. Standing hypotheses include (1) multiple timescale computations valuing the benefits of short-term uncertainty gains for long-term benefits as discussed above, (2) behavioral variability being induced by endogenous noise fluctuations in the brain [201], (3) specific brain regions which may seek to actively increase uncertainty or risk in some situations [202], such as the (poorly understood) circuits involved in curiosity, boredom and intrinsic motivation [106,130,171,203,204,205], or (4) responses to variations in the social environment and peer feedback [206,207,208]. Temporarily increasing local uncertainty is often necessary for generating alternative solutions in challenging environments and requires methods for amplifying internal or external sources of noise, inhibiting initial responses (to allow time to generate alternatives), and maintaining long-term goals [130]. Generally, there is an important need for developing multiscale learning and decision models that incorporate uncertainty (and mapping the neural correlates of these uncertainty mechanisms, e.g., the ACC [22]), as even variable learning rates do not fully address biological reactions to surprising outcomes for instance [25].

In addition, there are also other scales besides time and space, such as social scales. Decision making in social uncertainty (such as (4) above) is especially interesting, because social effects cannot be observed in typical behavioral experimental paradigms where only one isolated human is observed during an experiment [209]. However, social cues have been shown to be a dominant factor during exploration and foraging in many organisms [210,211], and to play a significant role in more effective and accurate human decision making [212,213].

Even in the growing area of environmental neuroscience, which encourages all experimental neuroscientists to consider the possible role of environmental factors (e.g., social factors, epigenetics, the physical environment, etc.) in influencing their results, a recent call was made urging researchers to consider hierarchical systems theory in experimental designs, as this multiscale theoretical framework has significant potential for studying how these more spatially and temporally broad social and environmental factors influence human behavior [20]. Due to the critical utility of multiscale processing, these topics will have a growing importance in future experimental and theoretical studies. 

## 7. Conclusions

In closing, multiscale processing is ubiquitous. It is necessary for foraging, visual processing, decision making, and problem solving, and both neuroscience and AI are making headway in understanding what biological and algorithmic mechanisms best achieve this. As we have shown, humans and other animals often appear to achieve this by modulating attentional resources across scales, with dedicated local and global processing architectures that are then integrated in specific brain regions to determine a final behavior. Researchers in AI are taking similar but also novel approaches, which allow systems to modulate across scales but also to process information in parallel formats (sometimes surpassing humans’ serial constraints, such as when humans read text). Ultimately, multiscale processing asks us to consider both short and long-term views and to choose in the context of changing environmental contexts. However, of course, the obvious question is what is the right mix of short and long scales, and how long is long enough? Biological systems may suffer under an evolutionary precedent to attend to multiscale events that only include near future and spatial (and social) scales, which bias decision making towards nearby short-term gains (and in-groups) [214]. There is no reason to think that AI will have similar constraints, unless we build them in. Thus, AI and neuroscience have much to offer one another, and there is still a great deal to learn.

## Figures and Tables

**Figure 1 brainsci-10-00396-f001:**
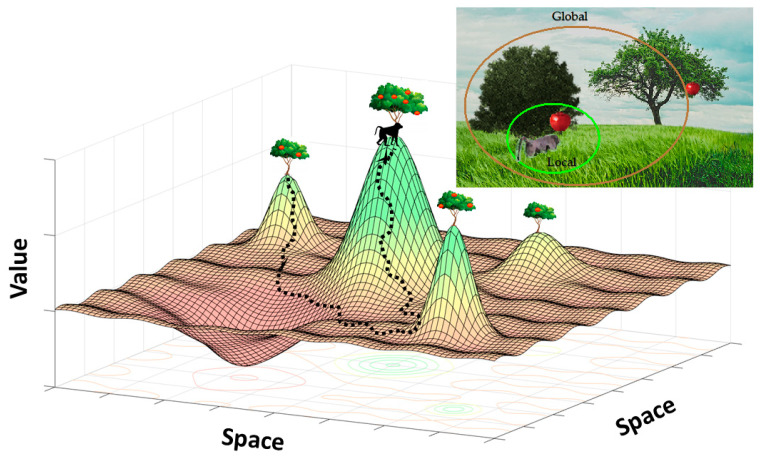
Environment Requiring Multiscale Processing. (Upper Right) For the animal at the tree in the front, the environment can consist of a local environment and a global environment. Its survival depends on its ability to process information at both local and global levels. For example, the larger environment may provide better opportunity for food sources and/or pathways for escape when predators appear. (Lower Left) The optimization process in machine learning and animals involves some sort of gradient ascent (or descent) where the state or agent follows the local gradient. This space can be considered continuous—Such as with “odor-scapes” of smell gradients—Or discrete—With different trees representing different patches, similar to multi-alternative choice environments (Upper right image adapted and modified with permission from Akaishi and Hayden, 2016, Neuron, Cell Press [35]).

**Figure 2 brainsci-10-00396-f002:**
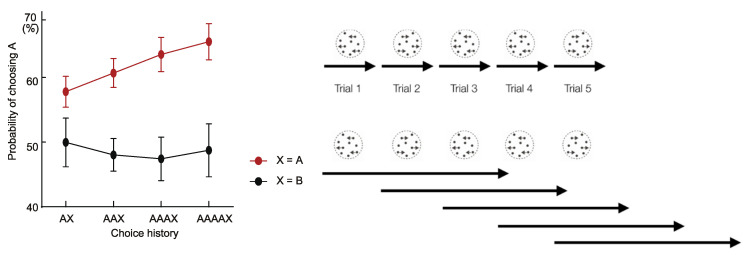
Behavioral Manifestations of Multiple Temporal Scales of Computations. The left panel shows the choice patterns across multiple trials. The *x*-axis indicates the number of repetitions of the same choice (A). The last trial (X), in the sequence AX, AAX, etc. on the independent axis in the figure, can be either A (repetitions of the same choices in the last trial) or B (the alternative in the last trial). The *y*-axis shows the probability of choosing A in the current trial. The right panel displays the scheme of multiple temporal scales in perceptual decision making, with two possible models. The black lines below the pictures of the stimuli of random dot motions indicate the effective length of computations. In the upper panels, the effective length is confined within a single trial (the model where each trial is independent), whereas the lengths in the lower panels are extended beyond the windows of single trials (the model where longer timescale choice bias exists). The plot on the left indicates that prior choices strongly bias future choices for the same task in the same environment, following the lower model. (Left image adapted with permission from Akaishi et al., 2014, Neuron, Cell Press [63]).

**Figure 3 brainsci-10-00396-f003:**
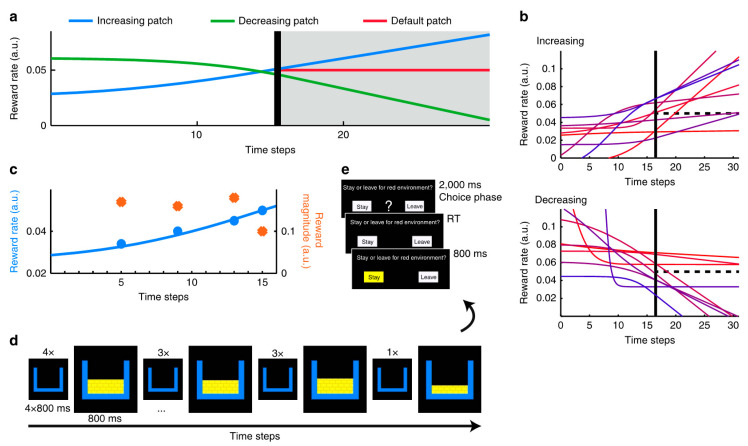
Patch Leaving Task used in the study by Wittmann et al. In this task, subjects decided whether to stay in the current patch or leave to another patch, relying on the experiences of past actions and reward outcomes. (**a**) Task flow. The *x*-axis is time steps, at which subjects take one action. The *y*-axis is an average reward rate. The subjects are either in the patches with gradually increasing reward rates (Increasing patch: the line in blue), or in the patches with decreasing reward rates (Decreasing patch: the line in green). The subjects take one action at each time step and receive a reward or non-reward. Then, after some time steps, they decided to stay or leave at the time predetermined by the experimenter (the thick vertical black line). After this patch leaving (staying) decision (the period right side of the thick vertical black line), the subjects continue the cycles of taking an action and receiving a reward (or non-reward). (**b**) Trajectories of average reward rates. There are several possible programmed trajectories for both increasing patch and decreasing patch. (**c**) Calculations of the average reward rates. The average reward rate is calculated by dividing an amount of the reward at each time step by a probability of reward occurrence at each time step. (**d**) Sequence of actions and rewards. There are cases in which a reward follows an action (yellow contents in buckets), or cases in which a reward does not follow an action (empty buckets). The amount of the content corresponds to the amount of the rewards. (**e**) Patch leaving decision. After the presentation of options (Stay or Leave), the subjects have to respond within 2000 milliseconds and the display of the chosen option appears 800 milliseconds after the decision response (Image adapted with permission from Wittmann et al., Nature Communications, 2016; Nature Publishing Group [66]).

**Figure 4 brainsci-10-00396-f004:**
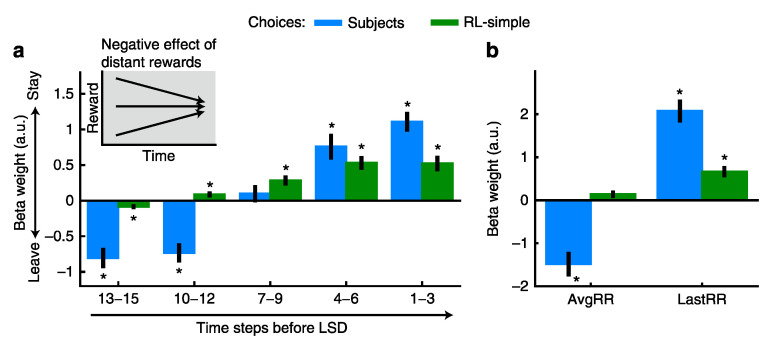
The weights assigned to the past reward experiences on the patch leaving decision in the study by Wittman et al. (**a**) The influences of the past rewarding events occurring in each past time periods on patch leaving decision. The *x*-axis indicates the past time periods (1–3, 4–6, 7–9, 10–12, 13–15: the lower numbers mean the proximity to the current time). The *y*-axis shows the weights on the patch leaving decision: positive values signifies influences for staying in the current environment and the negative values means the influences for leaving the current patch. Blue bars are the weights calculated from actual data of the subjects. The green bars are the weights calculated from the simulated data generated by the simple reinforcement learning (RL) model fitted to the actual subjects’ data. (**b**) The weights calculated for the short-term and the long-term reward rates on the patch leaving decision. The long-term (average) reward rates act to bias decisions for leaving the current environment and the short-term (last) reward rates influences decision for staying in the current patch (Image adapted with permission from Wittmann et al., Nature Communications, 2016; Nature Publishing Group [66]).

**Figure 5 brainsci-10-00396-f005:**
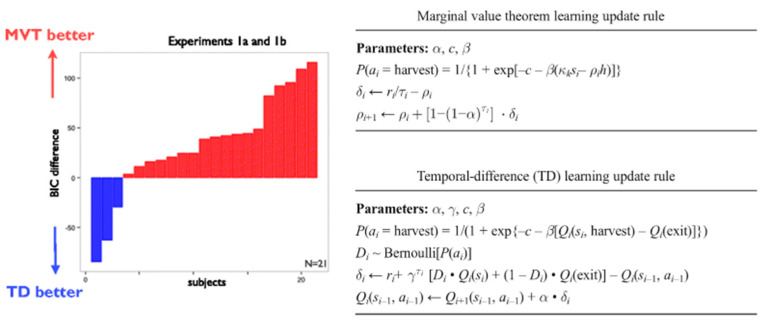
Model Comparison between Temporal Difference Learning and Marginal Value Theorem. (Left) The panel shows the Bayesian information criterion (BIC) result of the model comparison between the model of Marginal Value Theorem (MVT) and the model of temporal difference (TD) learning. The TD model is similar to the simple RL model in Figure 4. The red bars with values above zero refer to the subjects whose behaviors are better explained by the model of MVT, whereas the blue bars with negative values refer to the subjects whose behaviors are explained better by the model of temporal difference learning. (Right) These two tables summarize the MVT and TD models that were compared in the left plot. The MVT model optimizes undiscounted reward rate, whereas TD optimizes cumulative exponentially discounted reward with a free discount rate. For more detailed model information, see Constantino and Daw [95]. (Image adapted with permission from Constantino and Daw, Cognitive, Affective, & Behavioral Neuroscience, 2015; Springer [95]).

**Figure 6 brainsci-10-00396-f006:**
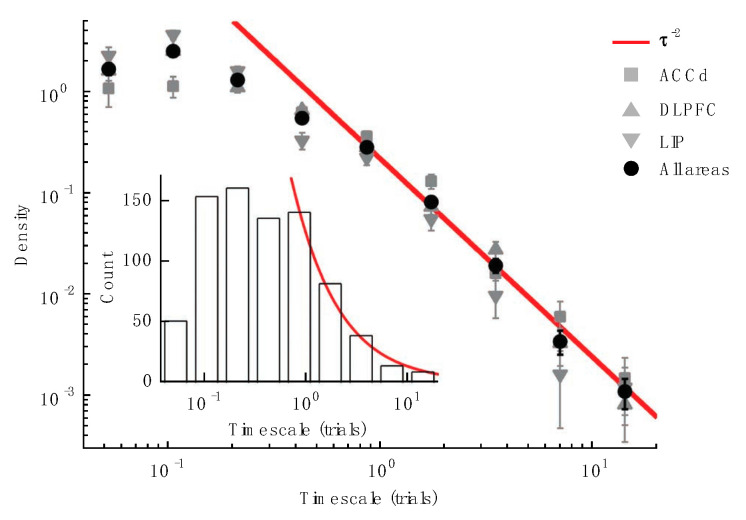
Multiscale Temporal Representations in Single Unit Activity. A figure panel from Bernacchia et al. (2011) The *x*-axis shows the time scale (in number of trials) in which past experiences can affect the firing rate. The *y*-axis indicates the probability density distribution of neurons having different time scales. The densities left of the point of 10^0^ = 1 trial shows the neurons with time scales shorter than a single trial, whereas the densities right of this point imply the neurons with time scales longer than one trial. There are substantial number of neurons showing the traces of memory carrying the information longer than a single trial (Image adapted with permission from Bernacchia et al., Nature Neuroscience, 2011; Nature Publishing Group [23]).

**Figure 8 brainsci-10-00396-f008:**
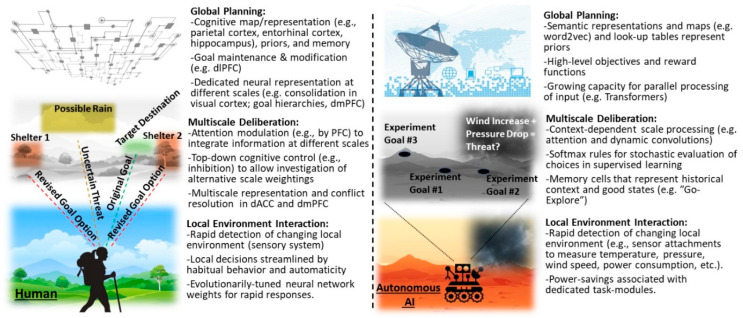
Comparing the Computational Architecture for Navigating Uncertain Environments of Human Brains and the Frontier of Autonomous AI Systems. (Left) Human brains have a highly flexible and generalizable learning architecture. Evolutionarily tuned biological networks allow for streamlining or even “automation” of local tasks and decision-making through biases and habituation, while cognitive maps and detailed prior knowledge within long-term memory allow for multiple goal tracking. Both top-down and bottom-up attentional processes modulate cognitive resources and deliberate across multiple spatio-temporal scales, while an agent pursues goals and makes inferences within dynamic and uncertain environments. (Right) Current AI systems have made tremendous progress towards comparable or even super-human computational and task-related performance in many controlled environments. AI already have the ability to simultaneously consider much larger amounts of sensory data and meta-data than human brains, with much larger-scale parallelization than biological brains, such as in Transformer networks. Thus, improved world models, more computationally efficient and robust multiscale attentional algorithms, and more flexible multiscale deep neural net structures that provide higher sample efficiency and accuracy hold great promise for achieving successful autonomous AI systems.

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
