# Peer review of "Multiscale Computation and Dynamic Attention in Biological and Artificial Intelligence"

_brainsci, 2020, doi:10.3390/brainsci10060396_

Round 1

Reviewer 1 Report

In this paper, the authors propose to examine complex cognition abilities in biological and artificial agents when facing multiscale (temporal and spatial) problems. In particular, the authors ‘focus on the struggle between mechanisms that seek to exploit known information and minimize risks associated with uncertainty, against ones that drive exploration and risk uncertain outcomes’, which is about the exploration-exploitation dilemma and seems at least partially disconnected/orthogonal to the issue of multiscale representations of time and space in the brain for efficient decision-making. The authors also discuss the way animals (and humans in particular) can flexibly alternate between different behavioral strategies when require, arguing that some strategies ‘can make computations more efficient by reducing short-term computational costs at the expense of long-term performance’.

The addressed issue is timely and important, because, as the authors argue in the discussion, ‘understanding the balance of these multiple competing and/or complementary computational layers underlying multiscale decision making (which may actually be the dominant form of decision making in humans) is a crucial problem for studies of information processing of the brain in the upcoming decade’ as well as to enable improved AI performance in the forthcoming years.

The paper is well written and covers a broad interdisciplinary literature which can be of broad interest for researchers in various fields. I do not have any major concern. I suggest below some corrections, rephrasing, and some relevant papers from the different research fields mentioned in the paper.

Minor points

In the first sentence of the introduction, I think it would be fair to also cite Tanaka, S. C. et al. (2004). Prediction of immediate and future rewards differentially recruits cortico-basal ganglia loops. Nature Neuroscience, 7(8).

Lines 48-49, some citations are needed in support of the following claim: ‘Biological cognitive systems often handle these problems by adaptively modulating attention across temporal and spatial scales.’ Do the authors include in this view the online decision to search deeper for old elements into memory as a function of uncertainty? The latter seems particularly relevant to me also to discuss the temporal processing paragraph, below, as example against the assumption that ‘the information processing in each trial is independent from information processing from other trials’. Do they further include online modulations of parameters of learning, which can produce a varying balance between exploration/exploitation (which is central in the argument of this paper) or the timescale of reward prediction,/learning?

The foraging problem, as framed here, very much resembles the multiscale spatial cognition problem. I think it would be particularly relevant to discuss computational models incorporating multiple spatial scales represented in hippocampal place cells.

About the temporal processing problem, there are striking examples of how information about past trials is encoded at the neural level and contributes to current decisions.

Lines 174-184: ‘Instead of always considering each option with equal distribution of time, the neural and behavioral patterns progressively show more fixed patterns and exploration of fewer alternatives. Thus this focusing of the mental processes on a fixed course of action makes the computation more efficient if the environment does not change drastically, but at the risk of future bias if the environment starts becoming more dynamic.’ Well, there is another possible interpretation coming from recent computational modeling work of Redish’s multiple T-maze task. The explanation is as following: if an agent learns this task with a model-free reinforcement learning augmented with replay mechanisms based on a buffer of episodic memory elements (note that such extension of classical model-free RL makes the agent’s behavior super flexible, in appearance nearly like model-based RL), then the more the agent learns the task, the more it focuses on the best option, thus the better the performance, and as a consequence the less its episodic memory contains elements representing alternative options. Consequently, mental replay processes will mostly focus on replaying the best option, and nearly not replaying alternative options. So, here the interpretation is that the mental focus on the best option is the consequence of a more focused behavior, which produces a more focused episodic memory, rather than the reverse.

In the study by Wittmann et al. 2016, whose detailed results are reported here and illustrated in Figures 3 and 4, I wouldn’t interpret a high reward rate in the long-term as having a negative influence of staying, because if the task had included a second increasing reward rate path with an offset (so that the initial reward rate is as high as the initial reward rate of the green path in Figure 3a, then I would expect this initial high reward rate to have a positive influence on staying. Note that this is the reason why the RL-simple model predicts a positive beta weight for time steps 10-12 in Figure 4a. My bet is that an average reward based RL model (i.e., a model where the reward prediction error is not computed simply based on the primary reward signal, but based on the difference of the primary reward signal and the estimated reward running average; see e.g., Palminteri, S. et al. (2015). Contextual modulation of value signals in reward and punishment learning. Nature communications, 6(1), 1-14.) might best explain these data. It is because the reward rate of the green path in the long-term is higher than average that it predicts that people want to leave, and because that of the blue line in the short-term is higher than average that it predicts that people want to stay. Would it be possible to re-analyse the data to see whether an average reward RL model is a better predictor of stay/leave than short-term/long-term?

The Temporal Difference learning model used by Constantino & Daw has not been described in the paper. So non-specialist readers might not be able to understand that this is exactly the same thing as the simple RL model used in Wittmann et al 2016. I think it would be useful if the authors could explain this in their manuscript.

Lines 446-448, ‘Thus further understanding of how humans are performing so well in these more complex environments, even in the face of sparse data, is a crucial factor in resolving the performance gap between human and AI agents in such tasks.’ This has long been the argument of researchers advocating bioinspiration in AI/robotics research.

Typos

References 59 and 112 are about the same paper.

Reference 81 lacks its author name: Charmov, E.L.

Line 207, ‘Marginal Value Theorem (Charnov 1976)[81].’. Please add (MVT) after Theorem, so that the reader quickly understands that MVT corresponds to this expression if he/she searches for the definition of this acronym in the paper.

Line 226, ‘to enhance our understand’ -> our understanding.

Line 253, ‘(Decreasing patch: the in green)’ -> the line in green.

Line 286, at the end of the title of Figure 4, please add ‘in the study by Wittmann et al.’.

Line 296, ‘staying the current patch’ -> staying in the current patch.

Line 293, ‘reinforcement learning model’ -> ‘reinforcement learning (RL) model’, so as to help the reader understand the caption in the figure.

Line 300, there are only two authors in the study of Constantino & Daw, so please remove ‘and his colleagues’.

Reviewer 2 Report

The authors present a review article that reports on biological and AI research on multiple temporal and special timescales.

Overall, the focus of the article is not entirely clear. Is this primarily a review of computational or brain models - or both? Is this more about the experimental designs or the underlying models? Does the plausibility of artificial intelligence mechanisms from a biological point of view play a role (e.g., the discussion on the plausibility of back propagation)?

Coming from a computer science background, I will focus more on the second part of the article. Regarding the first part: The studies were quite interesting to read - however, reviewers from the respective field have to judge if they present a good overview of the current scientific landscape in this field.

The presented work in the part on computational models is, in general, quite basic. The only algorithms that are discussed with regard to how they work are hill climbing (in the beginning) and convolutional neural networks. This might be suitable for a non-computer science audience, but it would have been more interesting to learn about the attention mechanisms in TransformerNet or the approaches used for RTS games.
Especially attention and self-attention only receive a very high-level explanation; how is it realized in terms of architecture? More importantly, why is this presented when the rest of the discussion is about Game AI for STR games? Transformer Net is used for language processing - how does this relate?

The biggest issue in this section is the selection of the application domain. Continuous-time and space have been tackled in many AI domains; however, the paper has a very narrow focus on game AI.
The recent literature on the use of continuous deep learning is, for instance, not reported entirely clearly (478: ... and even compared to the electronic Atari games played by DeepMind’s deep Q-network agent...) Here a deeper discussion might be good. It was one of the main achievements that the raw pixel input for the atari games was used for training - this is not different from the RTS games - only the output is different (joystick + button presses as opposed to mouse + key presses). Moreover, there are many extensions to deep reinforcement learning that also allow continuous actions.

In the discussion on RTS games, foraging is mentioned as a prominent factor; however, resource gathering is only one part of the game's challenge and usually not the most difficult one.

Overall, some high-level information on approaches is presented, but there is no systematic overview or categorization of approaches. The presented works seem arbitrary. This is even more unclear for the application domain; there are many application domains that deal with different time scales - why are RTS so prominent presented?

in summary, the computational part does not fulfill requirements of a review article. It is a rough overview of a few recent papers but requires considerably more depth.

Minor issues:
p2, l66: you speak about a "gradient descent process", but the figures show an ascent.

p3, l107: "two cases *in* that demonstrate"

p6, l226: enhance our understand*ing* of the

p12, l423 "pixel level classifications within objects[121]." --> within images

p13, l457: " with people’s" --> With

p15, l559 "understand" --> understanding

Author Response

Please see the attachment, this Reviewer's comments have been addressed in the "Reviewer 2" section, with the more neuroscience-focused Reviewer 1 comments also included because this Reviewer seemed to express interest in the neuroscience-focused critique of Reviewer 1.

Round 2

Reviewer 2 Report

Thank you very much for the revision. Your revised section on the computational models offers a broader overview that is more representative of the current state of the art in this field. Together with the changes to the introduction and the more clear target audience, the article is greatly improved and is a good contribution to the scientific community.

Below please find two minor issues that could be corrected before publishing:

line 533: LSTMs are also one type of recurrent neural network.

line 541: "By passing information through 540 kernels of varying size" - the kernels can also have the same size - what matters is that this a hierarchical process.